# The PRECISION study protocol: Can cervical stiffness in the second trimester predict preterm birth in high-risk singleton pregnancies? A feasibility, cohort study

Elizabeth Medford[1,2]*, Steven Lane[3], Andrew Sharp[1,2], Angharad Care[1,2]

1 Department of Women and Children's Health, Harris Preterm Birth Research Centre, University of Liverpool, Liverpool, United Kingdom, 2 Liverpool Women's Hospital, NHS Foundation Trust, Liverpool, United Kingdom, 3 University of Liverpool, Liverpool, United Kingdom

* emedford@liverpool.ac.uk

## Abstract

### Background

Preterm birth (PTB) is a leading cause of neonatal morbidity and mortality. More than 13 million babies are affected globally every year and PTB will contribute to over 900,000 deaths. In the UK, PTB affects 8% of pregnancies and costs more than £260 million annually in neonatal care. Identifying those at risk of this devastating complication of pregnancy and implementing preventative treatment remains a maternal health priority. The mainstay of PTB prevention has been assessment of cervical length (CL) in women at high-risk of PTB. However, CL has limitations, namely it is invasive, user dependent and varies over time through pregnancy. Importantly, not all those who are high-risk with a short CL will subsequently deliver preterm and CL screening doesn't identify a sub-set of women who have a spontaneous PTB without a short cervix. Therefore, existing care pathways for managing PTB can potentially benefit from additional assessments of risk. Novel ways of assessing cervical structure and function may improve our ability to predict spontaneous PTB and refine preventative intervention. This feasibility study will explore the use of a new antenatal test of cervical stiffness for assessing risk of spontaneous PTB in a high-risk singleton population.

### Methods

PRECISION is a single site prospective, feasibility, cohort study of asymptomatic women with singleton pregnancies at high risk for spontaneous PTB attending an inner-city tertiary maternity hospital in the UK. All study participants will be undergoing routine screening and management of PTB as per local guidance (NICE/Saving Babies Lives guidance) including CL screening with transvaginal ultrasound. Cervical stiffness will be assessed using the Pregnolia System; a novel, licensed, CE-marked, aspiration-based device. A measurement is obtained by applying the device directly to the anterior lip of the cervix, visualised via placement of a speculum, and gives a quantitative assessment of cervical

**Data availability statement:** Deidentified research data will be made publicly available when the study is completed and published.

**Funding:** Pregnolia AG provided the device for use in this study and provided financial aid to fund a Clinical Research Fellow to undertake the research. Pregnolia AG had no role in the study design, decision to publish or preparation of the manuscript.

**Competing interests:** Pregnolia AG provided the device for use in this study and provided financial aid to fund a Clinical Research Fellow to undertake the research. Pregnolia AG had no role in the study design, decision to publish or preparation of the manuscript. This does not alter our adherence to PLOS ONE policies on sharing data and materials

stiffness represented as the Cervical Stiffness Index (CSI, in mbar). Participants will undergo cervical stiffness assessments at up to three timepoints in the second trimester between $14^{+0}$ weeks and $25^{+6}$ weeks gestation. The cervical stiffness index data will be paired with routine PTB clinic CL measurements taken at the same time points. The primary outcome will focus on the feasibility of using this novel antenatal test in this high-risk population and explore any association between cervical stiffness and PTB.

## Discussion

This is an exploratory study to assess the use of this novel device in clinical practice. Direct comparison between cervical stiffness assessment using the Pregnolia System and CL assessment will determine the acceptability of this new assessment in this population, as well as explore its potential association with PTB. Our findings from this feasibility study will provide data on the potential of this novel device to impact PTB screening and evaluate acceptability of use in a high-risk population. Data on eligibility, recruitment rates and participant feedback will help inform future study design using the device.

## Trial registration

ClinicalTrials.gov NCT05837390

## Introduction

Preterm Birth (PTB), defined as delivery at less than 37 + 0 weeks gestation, is the single biggest cause of neonatal mortality and morbidity in the UK. Unfortunately it remains a common complication of pregnancy involving around 8% of births in England and Wales and affecting over 52,000 babies annually [1]. With increased surveillance, and intervention targeted towards preventing and managing PTB, mortality at extreme preterm gestations has improved. However, babies who survive PTB and are born at the earliest gestations suffer the greatest consequences of neurodevelopmental disability. Overall, the UK PTB rate has not declined over the last 10 years and the need for innovative evidence-based policy for PTB risk stratification remains a clinical priority [1–3].

Understanding the events leading to spontaneous PTB remains a significant research challenge due to its' complex and multifactorial aetiology [4,5]. Despite advances in PTB prediction and prevention a large proportion of spontaneous PTB remains undetected prior to the onset of labour or remains intractable to current interventions [4]. More than half of PTBs occur in women with no known or identifiable risk factors [6]. This reiterates the importance of developing novel approaches to predicting PTB.

Previous history of PTB is the strongest predictor of subsequent PTB and is used to define women as "high-risk" for PTB [7,8]. Transvaginal cervical length (CL) ultrasound assessment is widely accepted as the preferred screening tool used for predicting PTB in asymptomatic high-risk women [7,9]. These two findings form the basis of current PTB risk stratification in the UK through dedicated PTB clinics that perform surveillance second trimester CL scans on high-risk pregnant women. However, limitations exist with CL as a screening tool, and it remains unable to accurately predict all high risk women who will go on to have a spontaneous preterm birth < 34 weeks [10].

Premature cervical shortening precedes the onset of spontaneous PTB [7]. This is just one parameter of cervical remodelling that can be measured, as the cervix also needs to soften and dilate prior to spontaneous delivery [11]. Research into other bio-mechanical properties of

the cervix and its relationship to PTB have included cervical consistency and stiffness assessments using advanced ultrasound imaging techniques and mechanical tissue tests [4,6,12–15]. Importantly, it has been identified that the cervix softens before it shortens [13]. Pathological cervical remodelling can be detected earlier in asymptomatic individuals and with better diagnostic performance when measuring cervical stiffness compared to length for PTB prediction [13,16–18]. However, application of these proposed cervical consistency assessment tools relies upon expert ultrasound skills using expensive resources. So far, no objective technique to assess cervical softness during pregnancy has been well established in clinical practice [16,19].

The Pregnolia System is a CE-certified, novel device that provides objective, quantitative assessment of cervical stiffness through an aspiration-based technique [19,20]. Initial studies in a low-risk, unselected population have confirmed the device can detect cervical softening at an earlier gestation than ultrasound can detect cervical shortening, demonstrating its' potential use for PTB prediction [21]. Further studies have echoed this potential with findings of significant cervical softening in populations already identified as requiring intervention to prevent PTB [22,23]. So far, there has been no published evidence of using cervical stiffness (CSI, mbar) obtained with the Pregnolia System as a risk biomarker for PTB in an asymptomatic high-risk population. A feasibility study is needed to investigate the acceptability, implementation and efficacy of the Pregnolia System for cervical stiffness in the representative high-risk population to justify a further large scale predictive study and inform its appropriate design and power [24].

This feasibility study aims to confirm whether a lower cervical stiffness result obtained using the Pregnolia System has an association with spontaneous PTB in high-risk singleton pregnancies [25]. The study will also directly compare the current gold standard practice of serial CL surveillance in the second trimester to paired cervical stiffness assessments [1]. This study will explore how this device and its' results can be applied to the population served by the PTB clinic and review acceptability of this assessment in these patients. Together, these results will help inform a future study using the Pregnolia System as a potential predictive tool for PTB.

## Materials & methods

### Study design

This study is a single site prospective, observational, cohort study taking place at the Liverpool Women's Hospital in the UK. Participants will be recruited directly from the Preterm Birth Clinic (PTBC) at this tertiary women's hospital between $14^{+0}-25^{+6}$-weeks' gestation.

Participants will undergo cervical stiffness assessment using the Pregnolia System up to three times during their PTB surveillance $14^{+0}$ to $25^{+6}$ weeks gestation at study time points outlined in Fig 1. The CSI measurements will be undertaken alongside their CL scan measurements as part of their routine PTBC care [3].

Participants will complete a structured questionnaire after their first clinical examination in the study. This questionnaire will collect patient experience of the cervical stiffness assessment in comparison to standard CL assessment and collate data as to the acceptability of the assessment in this patient population (supporting information S1 Fig). Participants attending at Study Visit A time point will be invited to consent to gifting two high vaginal swabs as samples for future vaginal-microbiome research. These gifted samples will be obtained during the speculum procedure used for cervical stiffness assessment.

All study participation will coincide with the participants routine PTBC appointments, and participants will continue to receive all PTB intervention and surveillance as guided by their clinicians and as per national and regional PTB guidance (SBLCBV3, NICE Preterm Labour

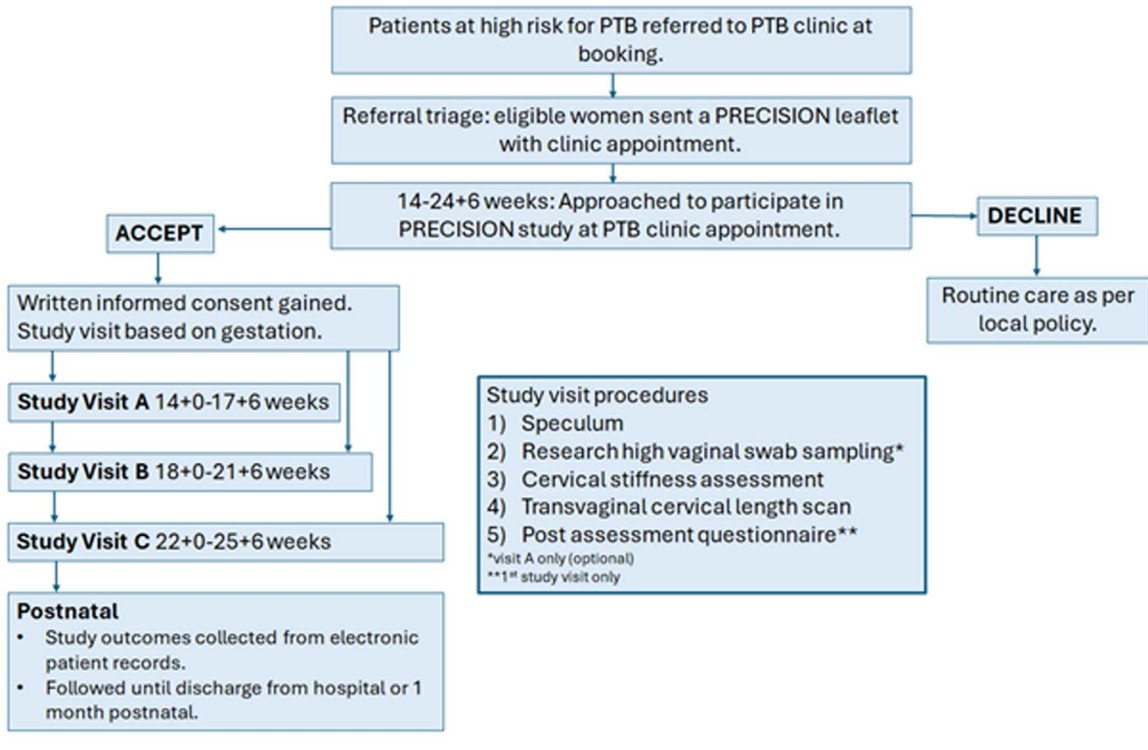

**Fig 1. Study Flow Chart.**

and Birth guidelines and the North West Regional network PTB guideline) [1,3,26]. Study procedures at each visit are detailed and explained in the study flow chart (Fig 1).

In this feasibility study we will recruit patients for 15 months and expect to achieve a cohort of 60 patients based upon expected high-risk clinic referrals and an assumed recruitment rate of 40%.

Participants will remain in the study following recruitment until delivery and discharge from hospital. Study outcomes will be collected from electronic hospital records for the participant and their baby.

## Study objective

The PRECISION study has the following objectives

- To obtain cervical stiffness measurements during the second trimester in high-risk singleton participants to:

  ◦ Determine the reliability and best interpretation of triplicate measurements in this patient group at this gestation.
  ◦ Explore any potential association between second trimester cervical stiffness and gestation at delivery.
  ◦ Compare cervical stiffness assessments to CL measurements taken concurrently and explore the association with gestation at delivery.

- To explore the acceptability of this assessment tool in this patient population for PTBC screening.

- Upon positive conclusion of the PRECISION feasibility study, to inform for the design of an appropriately powered study to assess the capability of this novel device for PTB prediction.

## Study population

The PRECISION study population will consist of consecutive pregnant women attending the PTBC at Liverpool Women's Hospital. Study eligibility will be determined at their initial attendance at the PTBC by the clinical team. Patient's meeting the inclusion and exclusion criteria, who wish to take part in the study must provide written informed consent for study procedures and use of their data from electronic medical records.

## Inclusion and exclusion criteria

All participants should meet the following inclusion criteria to be eligible:

- Age ≥ 18 years

- Singleton pregnancy

- Able to provide informed consent

- Meets criteria for high-risk PTBC, either:

  ◦ Previous preterm prelabour rupture of membranes (PPROM) < 34 + 0 weeks

  ◦ Previous PTB ≥ 16 + 0 to < 34 + 0 weeks.

Any possible exclusion criteria will be evaluated for eligibility for the study on the first visit and includes:

- Previous cervical surgery including previous trachelectomy, cone biopsy, loop excision or previous cerclage

- Existing cervical cerclage (vaginal or abdominal)

- Known cervical pathology at 12 o'clock position (cervical scarring due to prior large loop excision of the transformation zone (LLETZ) or cone biopsy/Nabothian cyst/polyp/cervical tears/cervical myomas/cervical condylomas/cervical endometriosis/cervical cancer)

- Vaginal bleeding evident on examination

- Visible, symptomatic cervical or vaginal infections

- Symptomatic of PTB (ruptured membranes, cervical dilatation, painful regular contractions)

- Known congenital uterine anomalies

- Known or suspected structural/chromosomal fetal abnormality

- Known HIV infection

## Study procedures

### Pregnolia System

The Pregnolia System is a novel medical device designed to quantitatively assess the biomechanical properties of the cervix through cervical stiffness. The system consists of a single-use sterile probe and a control unit as seen in Fig 2. The control unit is the active component that requires a power supply, foot switch for clinician control, and an integrated pump that generates a vacuum. The sterile probe is attached to the control device via a connector cable and air filters on the probe prevent microbiological contamination [27].

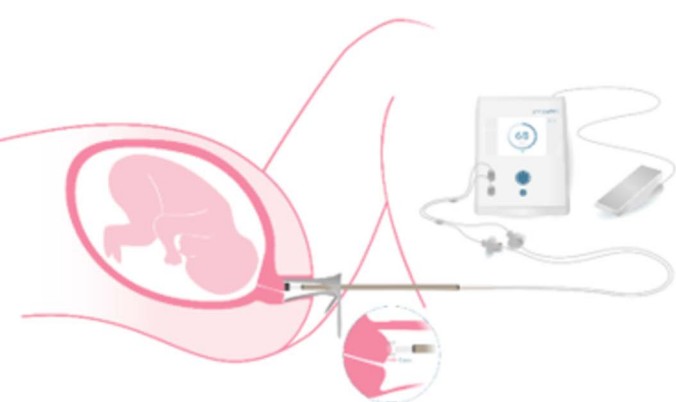

© Copyright 2024 Pregnolia AG. Reproduced with permission.

**Fig 2.  Pregnolia System-Control Unit with foot switch and single-use sterile probe.**

The device requires the use of a vaginal speculum to clearly visualise the cervix. The device probe is placed at the 12 o'clock position on the cervix and the cervical stiffness measurement is operated by a foot switch, creating a weak vacuum which displaces the cervical tissue into the probe tip to a depth of 4mm. The softer the tissue, the less pressure required to deform the tissue [28]. The result is represented by the CSI shown on the control unit. The Pregnolia system provides audio guidance to the clinician by indicating that the measurement has started, is in progress and is complete.

The device has clear operation instructions for use that outline appropriate clinical indications and contra-indications for use. All clinicians using the device will have read the instructions and completed the Pregnolia training instructional video.

## Cervical stiffness assessment

Cervical stiffness will be measured using the Pregnolia System. Following informed consent, all measurements will be performed with the woman in a supine position with an empty bladder. A vaginal speculum will be placed to visualise the cervix and the single-use, sterile Pregnolia Probe is placed on the anterior lip of the cervix at 12 o'clock position. A recording of cervical stiffness is generated over a maximum of 60 seconds (typically ~15 seconds) and displayed as CSI in mbar. The measurement is repeated 3 consecutive times without any time lag. The clinician is audio-guided by the control unit which indicates when a reading has been taken.

The cervical stiffness readings will be stored on the Pregnolia Control Unit and only documented for the participant once all study procedures have been completed during that study visit. The cervical stiffness readings are blinded to the patient and the clinician at the time of the study visit.

Any difficulty in obtaining readings will be documented on the case report form (CRF). Simple troubleshooting guidelines will be followed to optimize readings at the time of the study visit as per Pregnolia Instructions (e.g., avoiding kinks in connector cable and ensuring Luer locks are secure).

## Cervical length assessment

Clinical staff will perform transvaginal ultrasound assessment of CL using a GE Voluson E10 ultrasound machine and 7.5MHz transvaginal probe. All measurements will be performed with the woman in a supine position with an empty bladder in the sagittal plane measuring

from the internal to external cervical OS. The full length of the cervical canal will be visualized with the closed portion of cervix remaining being measured in triplicate.

The shortest CL measurement of the three readings will be documented as the final CL measurement for that study visit. CL measurements will be known to both the participant and clinician at the time of assessment and will be used to guide ongoing surveillance and PTB intervention as per the clinical team decision-making and regional guidelines.

The study procedure order is further outlined in the study flow chart (Fig 1) and schedule of study procedures (Table 1). Participants will remain in the study for a maximum of 31 weeks, from initial visit to PTBC until point of discharge after delivery, or infant aged 1 month. Assessments will take place at specified timepoints as shown in the study schedule (Table 1).

## Study outcome

The Pregnolia System is a novel device with no published data providing cervical stiffness index results in asymptomatic women who are high-risk for PTB. This will be the first study to provide this data and will guide how further research using this novel assessment tool can be most appropriately designed.

Previous studies at this study site over the last 10 years have achieved acceptable recruitment when obtaining blood samples and have demonstrated a notable PTB rate in a similar asymptomatic high-risk population (18%) [29,30]. However neither can be extrapolated to the PRECISION study due to the difference in study procedures and change in clinical practice over this time following the introduction of the Saving Babies Lives Care Bundle V3 [3]. Subsequently, this feasibility study will provide an accurate recruitment rate demonstrating this population's acceptability for this unique assessment in clinical practice. This study will provide an updated outcome incidence of PTB in this appropriately phenotyped study population and an estimate of the variance for cervical stiffness using the Pregnolia System. Together this data will infer the design of a larger study to establish a predictive model for spontaneous preterm birth (<34 weeks) in this patient population [31–33].

**Table 1. Schedule of Study Procedures.**

| Order of Procedures | At PTBC triage (12 weeks) | Study Visit A (14+0−17+6 weeks) | Study Visit B (18+0−21+6 weeks) | Study Visit C (22+0−25+6 weeks) | Postnatal & End of study |
|---|---|---|---|---|---|
| Patient Eligibility | X | | | | |
| Patient information Leaflet provided | X | | | | |
| Informed Consent | | X | | | |
| Pre-Sample Collection Questionnaire | | X | | | |
| High Vaginal Swab Collection (2 gifted samples) * optional | | X | | | |
| Cervical Stiffness Measurement | | X | X | X | |
| Cervical Length Scan | | X | X | X | |
| Post-Assessment Patient Questionnaire | | X | | | |
| Delivery Outcomes | | | | | X |
| Maternal Postnatal Outcomes | | | | | X |
| Neonatal Postnatal Outcomes | | | | | X |
| Adverse Event Reporting | | X | X | X | X |

## Feasibility outcomes

Our feasibility outcomes of interest relate to whether the study procedure is acceptable, participant adherence and retention to the study is achieved, data collection is feasible, and that cervical stiffness assessment fidelity is maintained with adequate reliability and safety[34].

Our feasibility outcomes will be defined as follows;

1) Participant acceptability of cervical stiffness assessment in PTB clinic

    a. Qualitative questionnaire following assessment
    b. Number of participant withdrawals throughout the study duration (aim for < 20%)
    c. Reasons for non-participation collected of eligible women approached for the study

2) Adherence to the protocol

    a. Measured as number of protocol deviations

3) Adherence to data collection

    a. Numbers of missing data (aim for < 10%).
    b. Loss to short term follow up (aim for < 5%)

4) Safety outcomes

    a. Incidence of adverse events

5) Cervical stiffness fidelity

    a. Ability to obtain triplicate cervical stiffness measurements (aim for > 90%)
    b. Triplicate measurement reliability (aim for > 0.8 Cronbach's Alpha)

6) Inputs for sample size and power calculation of future study

    a. Determine sPTB rate in the total study population (participating and non-participating PTBC patients) to inform current sPTB rate post national guidance care implementation
    b. Determine estimate of CSI variance using the Pregnolia System
    c. Determine recruitment rate to the study to inform sample size required

## Clinical outcomes

Clinical outcome data will be collected in accordance with the international clinical Delphi consensus on the core outcome data set for PTB in asymptomatic women [35]. The primary clinical outcome is PTB < 34 + 0 weeks gestation. We will define PTB as labour having occurred with either intact membranes or PPROM (<37 + 0 weeks). This excludes iatrogenic causes for preterm delivery including induction of labour (in the absence of PPROM) or elective caesarean section. Secondary maternal and neonatal outcomes will be collected for descriptive analyses only.

## Study endpoint

The study will end when the last recruited woman has delivered and both herself and her baby have been discharged from hospital, or 1 month after delivery, whichever is sooner and all planned analysis of collected data has taken place.

## Statistical analysis

Descriptive statistics will be generated and presented as means (SD), median (IQR) and frequency of observations (percentages) with 95% confidence intervals as appropriate.

Cronbach's Alpha will be used to assess the reliability of the device to produce consistent readings at different time points. It compares shared variability among the measurements with the overall variance. If the device is reliable then there should be high covariance amongst the measurements relative to the variance. Cronbach's alpha alongside descriptive statistics with 95% confidence intervals will inform best use of the triplicate CSI measurements taken at each study visit. Specifically exploring whether the first, average, median or lowest measurement of the three readings should be utilised in further analysis and most importantly inform best use of the CSI results in clinical practice.

An estimate of the variability in the outcome measure will be calculated and then applied to a sample size calculation for a further definitive study.

Diagnostic performance of cervical stiffness assessments using the Pregnolia System for PTB will be demonstrated through receiver operating characteristic curves with area under the curve and 95% confidence intervals being calculated, as well as aiming to define the optimum cut-off value for predicting PTB. Cervical length diagnostic performance data will also be generated for direct comparison.

Any association between cervical stiffness and gestational age at delivery will be explored using scatter plots and linear regression. Longitudinal data will be explored using time series plots, and box plots with a sensitivity analysis completed on all participants with a complete data set.

## Data management plan

All study participants are allocated a unique participant identification number and all data relating to that patient is pseudonymised. Study data is initially captured using a paper CRF. The data from the paper CRFs are then transcribed to an electronic CRF within a bespoke, password protected, study database (REDCAP). The Chief Investigator (CI) will preserve the confidentiality of participants taking part in the study and will abide by the EU General Data Protection Regulation 2016 and Data Protection Act 2018.

### Consent and criteria for withdrawal

Consent to enter the study must be sought from each participant only after a full explanation has been given, an information leaflet offered, and time allowed for consideration. Signed participant consent will be obtained. The participant can decline to participate without giving reasons and this will not impact upon further care. All participants are free to withdraw at any time from the study without giving reasons and without prejudicing further care.

In addition, the CI may decide, for reasons of medical prudence, to withdraw a participant. In either event, the Sponsor will be notified and the date and reason(s) for the withdrawal will be documented in the participant source data. If a participant withdraws or is withdrawn, ideally, they should remain in the study for the collection of outcome data. If the participant states their wish not to contribute further data to the study, collected data will be removed from the study database and no further outcome data will be collected.

### Monitoring and safety

A Study Management Group (SMG) comprising the CI, principal investigator, co-applicants, and core study management staff meet at regular intervals throughout the course of the study and holds responsibility for the day-to-day running and management of the study. The need to stop the study will be determined by the SMG and the decision will be based upon data integrity and participant safety.

All adverse events for this study are recorded at each study visit on the study CRF. This is a non-interventional study we therefore do not anticipate many serious adverse events. The Pregnolia System is a licensed, CE-marked, non-invasive medical device and the risk of adverse events related to the measurement is small. The device has a very good safety profile, and the manufacturer has not received any serious adverse event reporting related to the medical device either in studies or clinical practice at the time of starting the PRECISION study.

### Ethical considerations and declarations

The Wales Research Ethics Committee 3 has given approval for this research (23/WA/0044). All patients will be given written informed consent prior to entry to the study and will be aware that participation is completely voluntary.

### Status and timeline of the study

PRECISION has been open for recruitment since 6th July 2023. Recruitment is due to finish in October 2024, and patient follow up complete in January 2025. The study will end at 20 months from opening which includes all recruitment, follow up and analysis of data.

## Discussion

The PRECISION study aims to evaluate the use of cervical stiffness as a novel assessment tool for PTB risk prediction in high-risk asymptomatic singleton pregnancies. Identification of a new effective screening tool that can either work independently or alongside current screening tools may hold significant potential to improve PTB management. Improved identification of those at risk could allow for earlier PTB interventions and effective PTB optimisation bundles to be appropriately implemented.

To date there have been no clinical studies of the Pregnolia System performed in a high-risk pregnancy population for the prediction of spontaneous PTB. With the resource constraints of research, not all interventions can be tested for both efficacy and effectiveness. Feasibility studies are recommended to establish which tools should be recommended for efficacy testing [24]. Although other research studies are currently underway to attempt to assess this tool [36], we have designed a feasibility study to firstly assess recruitment rates, acceptability in practice and identify the likelihood of clinical effectiveness prior to designing and costing an appropriately powered study to attempt to demonstrate efficacy of this tool. We also wish to inform our analysis on how best to use the triplicate readings recommended by Pregnolia in preparation for an appropriately designed larger study in our population that can then fully explore the device's potential predictive capacity for PTB.

Although this feasibility study is limited by its single centre design, its strengthened by its' setting in an inner-city tertiary maternity hospital that serves a diverse population with a known high PTB rate. The pragmatic study design allows all eligible participants who attend the PTB service at the study site to participate and captures use of the assessment tool in real clinical practice. The clearly defined study population will allow the results from this feasibility study to be truly applicable to the population a future clinical prediction model would serve.

The PRECISION study will provide unique data of cervical stiffness in high-risk asymptomatic singleton pregnancies that is not yet available in the literature. This novel clinical data will be instrumental to inform a larger study that has the potential to implement important change in such a high-risk cohort of obstetrics.

## Supporting information

**S1 Fig. Participant Questionnaire.**
(TIF)

**S1 File. Precision ICF.**
(PDF)

**S2 File. Precision PIS.**
(PDF)

## Author contributions

**Conceptualization:** Elizabeth Medford, Andrew Sharp, Angharad Care.

**Formal analysis:** Elizabeth Medford, Steven Lane.

**Funding acquisition:** Angharad Care.

**Investigation:** Elizabeth Medford.

**Methodology:** Elizabeth Medford, Steven Lane.

**Project administration:** Elizabeth Medford.

**Supervision:** Andrew Sharp, Angharad Care.

**Visualization:** Elizabeth Medford.

**Writing – original draft:** Elizabeth Medford.

**Writing – review & editing:** Steven Lane, Andrew Sharp, Angharad Care.

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
