## [Decision Letter · Decision Letter 0]

25 Jun 2024

PONE-D-24-18823Study Protocol. The PRECISION study protocol. Can cervical stiffness in the second trimester predict preterm birth in high-risk singleton pregnancies? An exploratory, prospective cohort study.PLOS ONE

Dear Dr. Medford,

Thank you for submitting your manuscript to PLOS ONE. After careful consideration, we feel that it has merit but does not fully meet PLOS ONE’s publication criteria as it currently stands. Therefore, we invite you to submit a revised version of the manuscript that addresses the points raised during the review process.

**ACADEMIC EDITOR: Please respond to all reviewers comments**==============================

We look forward to receiving your revised manuscript.

Kind regards,

Ahmed Mohamed Maged, MD

Academic Editor

PLOS ONE

Journal Requirements:

2. Thank you for stating the following in the Acknowledgments Section of your manuscript: " Pregnolia AG provided the device for use in this study and provided financial aid to fund a Clinical Research Fellow to undertake the research. Pregnolia AG and the study sponsor had no role in the study design, decision to publish or preparation of the manuscript. "

Please remove any funding-related text from the manuscript and let us know how you would like to update your Funding Statement. Currently, your Funding Statement reads as follows: "Pregnolia AG provided the device for use in this study and provided financial aid to fund a Clinical Research Fellow to undertake the research.  Pregnolia AG had no role in the study design, decision to publish or preparation of the manuscript. "

3. Thank you for stating the following in the Competing Interests section: "Pregnolia AG provided the device for use in this study and provided financial aid to fund a Clinical Research Fellow to undertake the research.  Pregnolia AG had no role in the study design, decision to publish or preparation of the manuscript.  " 

Reviewers' comments:

Reviewer's Responses to Questions

**Comments to the Author**

1. Does the manuscript provide a valid rationale for the proposed study, with clearly identified and justified research questions?

Reviewer #1: Yes

Reviewer #2: Partly

2. Is the protocol technically sound and planned in a manner that will lead to a meaningful outcome and allow testing the stated hypotheses?

Reviewer #1: No

Reviewer #2: Partly

3. Is the methodology feasible and described in sufficient detail to allow the work to be replicable?

Reviewer #1: No

Reviewer #2: No

4. Have the authors described where all data underlying the findings will be made available when the study is complete?

Reviewer #1: Yes

Reviewer #2: Yes

5. Is the manuscript presented in an intelligible fashion and written in standard English?

Reviewer #1: Yes

Reviewer #2: No

6. Review Comments to the Author

You may also provide optional suggestions and comments to authors that they might find helpful in planning their study.

Reviewer #1: The methodology and findings of the study is very limited. Sample size is very small for developing a prediction model. Even though t test or chis-square test are not enough methodology for a prediction model. At least the odds ratio of a univariate or multivariable logistic would be helpful. I would refer to author to see (and cite) the methodology of the following papers.

https://pubmed.ncbi.nlm.nih.gov/34191801/

https://pubmed.ncbi.nlm.nih.gov/34112965/

Reviewer #2: SUMMARY

In their publication on the PRECISION study protocol, Medford et al outline a single-center prospective cohort study involving pregnant women identified to be at high risk of spontaneous preterm birth which aims to evaluate cervical stiffness as a predictor of preterm birth at different 3 timepoints in pregnancy. Cervical Stiffness is measured using a novel medical device, “Pregnolia” which measures the vacuum as required to deform cervix tissue to a certain extent. The vacuum required to obtain the deformation provides a measure of cervical stiffness (high vacuum required) or cervical softening (low vacuum required), with the latter indicative for cervical remodelling preceding cervical shortening and possible preterm labour. In addition, the authors want to compare Cervical Stiffness with Cervical Length and quantitative fetal fibronectin analyses for Preterm birth (before 34 weeks of gestation) prediction. In addition, the PRECISION study will collect patient feedback on how they experience the three PTB risk assessment procedures. The authors also want to explore whether combining the different preterm birth risk assessments will improve PTB prediction.

MAJOR ISSUES

1) The authors decided upon a patient recruitment target of n = 60 for the PRECISION study without a rationalization (sample size calculation) of this number in view of the study objective(s), i.e., the use of Cervical Stiffness (as assessed with the Pregnolia device) as a predictor for preterm birth in an (asymptomatic) high risk patient population. The authors justify their choice of not having performed a power analysis due to the Pregnolia device being a novel device with limited clinical data.

It is noted that Breuking et al recently published a study protocol which targets the same patient population and has largely the same aims as the PRECISION study [Cervical Stiffness as Assessment of cervical softening and the prediction of preterm birth (STIPP): protocol for a prospective cohort study https://bmjopen.bmj.com/content/bmjopen/13/11/e071597.full.pdf].

Different from the PRECISION protocol authors, Breuking et al provided a sample size estimation which suggests a minimal sample size of n=227 to evaluate Cervical Stiffness as a predictor for spontaneous preterm birth. Given the high degree of similarity in research question and research context on the one hand, yet the largely divergent recruitment targets, it is advised to verify (non-)futility of PRECISION by performing a formal sample size calculation or provision of another justification of the sample size. With Breuking et al in the public domain, the authors may want to replicate the power analysis as conducted by Breuking et al, using local estimates of prevalence etc, Based on their findings, the authors may possibly want to adjust the study aims accordingly as per the option to amend the protocol as per the language in the first paragraph of the “Introduction” on page 11 of the Precision Study Protocol.

In addition, given the apparent similarities between the PRECISION and STIPP (more specifically, the A-STIPP sub-study), the authors should reference Breuking et al and discuss similarities and differences between the studies.

MINOR ISSUES

1. Abstract

1.1. Background

- To frame the PRECISION study, the authors identify that Cervical Length assessment, the best [current] available test for PTB prediction in a high-risk [pregnancy] population, has limited value in PTB risk prediction in the low-risk population. Given that the authors identify the lack of adequate PTB prediction in the low-risk population as the most pressing unmet clinical need, the background section would benefit of a justification/rationale why the results of the PRECISION study, as obtained in a high-risk population, would help in addressing the unmet clinical need in the low-risk population.

1.2. Methods

- Line 29: remove “non-invasive”

1.3. Discussion

- Line 41: contrast cervical stiffness as a novel screening tool with the current screening tools (CL, fFN).

- Line 42: the authors may want to re-evaluate whether exploration of combining cervical stiffness, CL and fFN, into a multivariable model (cf. synergistic potential) is possible within the sample size.

2. Introduction:

- Lines 50-59: In the first paragraph, the authors are providing some background to the PRECISION study. For the non-PTB researcher, the following would be helpful:

-reiteration of the clinical need

-the reasons why effective PTB risk prediction remains elusive

-mention of other predictors than cervical remodelling parameters like intrauterine infection and the role of FFN

-clarification that currently cervical remodelling assessment is based on sonographic assessments: CL (standard

care) and cervical consistency Index (subject to research)

-better explain why CCI has limitations (like?). It is noted that this statement appears at odds with the

statement made on page 13 of the Clinical Protocol, which mentions that CCI is an “easily, reproducible

assessment and effective in the prediction of spontaneous PTB”.

- Line 66: The predictor under investigation is better defined as the Cervical Stiffness as obtained with the Pregnolia device and expressed as the CSI (mbar).

- Line 66: Within the context of PRECISION, it is unclear whether CSI is evaluated as a diagnostic biomarker (to confirm preterm labour; if yes – discussion should discuss comparison with QUIPP app decision support tool) or as a risk biomarker; it would be helpful if the authors can clarify this ambiguity, f.i, using the publicly available FDA-NIH resource on different types of biomarkers [https://www.ncbi.nlm.nih.gov/books/NBK338449/]

- Lines 70-71: add reference to clinical guidance [NICE?] containing current gold standard practice as referred to

- Lines 70-73: The authors appear to suggest the use of longitudinal data for comparing CSI with CL and fFN; it is noted that such analyses are not mentioned in the statistical analysis?

- Lines 73-74: As it stands, it is unclear which and how data from the PRECISION study will be used to inform a future clinical trial. It would be helpful if the authors can add this detail in the Discussion.

- Note: The Background Section 1.1 in the actual Study Protocol is very well organized and clearly presented. The authors may consider replicating the same background presentation sequence (albeit condensed)

3. Materials & Methods

3.1. Study Design

- Line 79: The language suggests that recruitment occurs at first clinical appointment which happens between 14+0

– 25+6 weeks? Please clarify that recruitment is at first clinical appointment in the PTBC [14+0 – 17+6 weeks],

with 2 additional CS analyses scheduled as part of routine appointments scheduled before 25+6 weeks?

It is noted that the 3rd text box from the top in Figure 1 contains the same ambiguity.

- Line 84: If possible, reference guidance which outlines routine PTBC care

- Line 85: clarify that the questionnaire is only offered post first CL assessment.

3.2. Study Objective

- Line 106: the language in objective 1.3 is unclear, can the authors clarify what is meant with “… explore the relative association with gestation at delivery”.

- Line 109: Objective 3 – details on how this study will inform a next study would be helpful; see higher.

3.3 Inclusion and Exclusion Criteria

- OK

3.4. Sample Size Calculation

- Refer to Major Issue #1. It is noted that the authors have estimates on # referrals, # patients meeting eligibility, recruitment rate. If the authors can generate an estimate of the prevalence of sPTB in their Preterm Birth Clinic, they should be well equipped to conduct a power calculation. At a minimum, the latter prevalence estimate would allow for estimating the number of expected SPTB<34 weeks and application of the crude rule-of-thumb of having at least n=10 cases (or controls) per variable considered in a multivariable analysis.

3.5. Study Procedures

3.5.1. Pregnolia System

- OK

3.5.2. Cervical Stiffness Assessment

- Lines 175 – 177: Can the authors elaborate on how the clinician will be blinded to the CSI as the schematic in Figure 2 and the language (line 163) suggests that the CSI is displayed on the control unit when performing the measurements

3.5.3. Cervical Length Assessment

- OK

3.5.4. Fetal Fibronectin Assessment

- Lines 193 - 200: Content of this first paragraph is better presented in the introduction

- Lines 198 – 200: Based on the outcome of above looked-for sample size calculation and /or estimation of sPTB <34 weeks expected, the authors are advised to re-evaluate whether this objective is meaningful within the n = 60 context of PRECISION

3.5.5. Post Assessment Questionnaire

- Lines 215 – 218: This content is more appropriately presented under Section Materials Methods – Study Design.

3.5.6. Study Outcome

-Lines 222 – 223: Use of “definitive” language can improve readability, e.g: The primary outcome is…. ; sPTB is defined as…

3.5.7. Study Endpoint

- OK

3.5.8. Statistical Analysis

- This section would benefit from a more structured organisation, fi,

- Descriptive statistics will be generated and presented as means (SD), median (interquartile range [IQR]), and

frequency of observations (percentages), as appropriate. Comparisons of patient characteristics between

women with and without the primary outcome sPTB<34 weeks will be performed using chi-square

(categorical data) or student-t test or Mann−Whitney U tests, as appropriate following normality testing

(Shapiro-Wilk).

[Note, the study’s n is sufficient to perform chi-square tests, there is no need of Fisher’s exact test]

- If there are >= 10 events,… logistic regression and ROC cure analysis will be performed to evaluate the

prediction performance of the predictors CS, CL and FFN… ;

Note: meaningful cut-offs are typically informed by clinical utility and typically require for selecting a criterion,

fi. fixed False Positive Rate = XX%; given the exploratory nature of PRECISION, the authors may consider this

as an objective for a follow-up study.

- [Multivariable analysis will be performed using Log Reg …etc;

Note: The authors are advised to re-evaluate whether this analysis is meaningful within the n = 60 context of

PRECISION]

-A p<0,XX will be used as the significance level in all statistical analyses

- Note: In other places, the author appear to indicate that prediction analysis will involve the use of longitudinal data as well. If this is the case, then the authors should update the Statistical Analysis Section with a description of what prediction modelling methods will be applied.

3.5.9. Data Management Plan

- With recruitment already started, use of definitive language may be more appropriate in this section

3.5.10. Consent and Criteria for Withdrawal

- OK

3.5.11. Monitoring and Safety

- With recruitment already started, use of definitive language may be more appropriate in this section

- Line 278: it may be appropriate to qualify the Pregnolia SAE reporting to e.g. a certain time point, like no SAE were reported at time of starting the PRECISION study.

3.5.12. Ethical considerations and Declaration

- OK

3.5.13. Status and timeline of the study

- OK, refer to Major Issue Section regarding (non-)futility of the PRECISION study

4. Discussion

- Lines 301-302: The authors suggest that the effect of PTB interventions on CSI will be captured. There is however no mention of this in the Study Objectives nor in the Statistical Analysis Section. It would be helpful to the reader to understand how the authors will assess such intervention effects as part of the Statistical Analysis.

- Lines 303-307: The authors should add some commentary regarding the published study protocol of Breuking et al vs the PRECISION protocol.

5. Author’s Contributions

- OK

6. Acknowledgments

- It is noted that the funding by Pregnolia is not disclosed in the official Study Protocol; Section 7.9.

7. PLOS authors have the option to publish the peer review history of their article (what does this mean? ). If published, this will include your full peer review and any attached files.

**Do you want your identity to be public for this peer review?** For information about this choice, including consent withdrawal, please see our Privacy Policy .

Reviewer #1: **Yes: ** Reza Arabi Belaghi

Reviewer #2: No

---

## [Author Response · Author response to Decision Letter 1]

12 Aug 2024

Editors comments.

Author response: I believe my manuscript meets the PLOS ONE style requirements.

2. Thank you for stating the following in the Acknowledgments Section of your manuscript: " Pregnolia AG provided the device for use in this study and provided financial aid to fund a Clinical Research Fellow to undertake the research. Pregnolia AG and the study sponsor had no role in the study design, decision to publish or preparation of the manuscript. "

Please remove any funding-related text from the manuscript and let us know how you would like to update your Funding Statement. Currently, your Funding Statement reads as follows: "Pregnolia AG provided the device for use in this study and provided financial aid to fund a Clinical Research Fellow to undertake the research. Pregnolia AG had no role in the study design, decision to publish or preparation of the manuscript. "

Author response: I would like to confirm that I have removed the funding information from my manuscript. My funding statement remains correct and does not need amending.

3. Thank you for stating the following in the Competing Interests section: "Pregnolia AG provided the device for use in this study and provided financial aid to fund a Clinical Research Fellow to undertake the research. Pregnolia AG had no role in the study design, decision to publish or preparation of the manuscript. "

Author response: My competing interest statement is as follows:

Pregnolia AG provided the device for use in this study and provided financial aid to fund a Clinical Research Fellow to undertake the research. Pregnolia AG had no role in the study design, decision to publish or preparation of the manuscript. This does not alter our adherence to PLOS ONE policies on sharing data and materials.

Author response: I can confirm that I have updated a Data Availability Statement in the submission form.

Author response: I have amended my ethics statement to appear only in my methods section.

Reviewers comments.

Major issues

1) The authors decided upon a patient recruitment target of n = 60 for the PRECISION study without a rationalization (sample size calculation) of this number in view of the study objective(s), i.e., the use of Cervical Stiffness (as assessed with the Pregnolia device) as a predictor for preterm birth in an (asymptomatic) high risk patient population. The authors justify their choice of not having performed a power analysis due to the Pregnolia device being a novel device with limited clinical data.

It is noted that Breuking et al recently published a study protocol which targets the same patient population and has largely the same aims as the PRECISION study [Cervical Stiffness as Assessment of cervical softening and the prediction of preterm birth (STIPP): protocol for a prospective cohort study https://bmjopen.bmj.com/content/bmjopen/13/11/e071597.full.pdf].

Different from the PRECISION protocol authors, Breuking et al provided a sample size estimation which suggests a minimal sample size of n=227 to evaluate Cervical Stiffness as a predictor for spontaneous preterm birth. Given the high degree of similarity in research question and research context on the one hand, yet the largely divergent recruitment targets, it is advised to verify (non-)futility of PRECISION by performing a formal sample size calculation or provision of another justification of the sample size. With Breuking et al in the public domain, the authors may want to replicate the power analysis as conducted by Breuking et al, using local estimates of prevalence etc, Based on their findings, the authors may possibly want to adjust the study aims accordingly as per the option to amend the protocol as per the language in the first paragraph of the “Introduction” on page 11 of the Precision Study Protocol.

In addition, given the apparent similarities between the PRECISION and STIPP (more specifically, the A-STIPP sub-study), the authors should reference Breuking et al and discuss similarities and differences between the studies.

Author response: To provide a power calculation to assess if cervical stiffness is an appropriate tool for use in clinical practice several pieces of information are required, including a clear research question. Although the expected prevalence of sPTB <34 weeks is known from the high-risk population that is being sampled, it is not known what the prevalence of disease is amongst women consenting to inclusion. These women have all had a previous spontaneous preterm birth or late miscarriage and may not wish to have any invasive procedures, particularly involving or affecting the cervix. Breuking et al have used a standard level of variance for their calculation, however there is no clinical data yet available to show that the Pregnolia device can achieve this level of variance in this population. There is also an assumption that this tool will provide accurate readings in each recruited individual, despite never having been tested in this population. There may be a training period required for the user to get used to the equipment, resulting in unreliable readings. The manufacturers recommend taking triplicate measurements of cervical stiffness, but do not offer any guidance on which of these three measurements or how these measurements should be used in analysis. It is also unclear how one measurement will affect the following measurements given that the sampling point on the cervix is the same. We are performing a feasibility study recruiting over 15 months to address some of these questions and to determine if this is an intervention which is appropriate for further testing. Results will facilitate an informed sample size calculation for such a study and study design, if appropriate.

Our proposed figure of recruitment of 60 patients in our study period of 15 months was a conservative estimate based on our ability to recruit to other biomarker studies in the same population (ref Gupta JK, Care A, Goodfellow L, Alfirevic Z, Müller-Myhsok B, Alfirevic A. Genome and transcriptome profiling of spontaneous preterm birth phenotypes. Sci Rep. 2022 Jan 19;12(1):1003. doi: 10.1038/s41598-022-04881-0. Erratum in: Sci Rep. 2022 Feb 1;12(1):1986. doi: 10.1038/s41598-022-06338-w. PMID: 35046466; PMCID: PMC8770724.). Based upon the biomarkers study, our preterm birth rate in this population is estimated at 18% and therefore we might expect 10 preterm births in 60 recruits. If this number is achieved, we may be able to perform some basic prediction analysis of the test with logistic regression assessing just one predictive variable. However, this is not the purpose of this feasibility research study, and we recognise that overestimation in event rates are common when data is used in this way (ref Oliver CB, Strub L, Sunnen et al. Accuracy of Event Rates and Effect Size in Estimation in Major Cardiovascular Trials: A systematic Review. Jama Netw Open 2024; 7(4): e248818. Doi: 10.1001/jamanetworkopen.2024.8818)

MINOR ISSUES

1. Abstract

1.1. Background

- To frame the PRECISION study, the authors identify that Cervical Length assessment, the best [current] available test for PTB prediction in a high-risk [pregnancy] population, has limited value in PTB risk prediction in the low-risk population. Given that the authors identify the lack of adequate PTB prediction in the low-risk population as the most pressing unmet clinical need, the background section would benefit of a justification/rationale why the results of the PRECISION study, as obtained in a high-risk population, would help in addressing the unmet clinical need in the low-risk population.

Author response:

Predictive tools are affected by prevalence of disease. As the prevalence increases the PPV increases and the NPV decreases. In the low-risk population, the prevalence of preterm birth is naturally much lower. However, if this test shows no predictive value in the high-risk population, it almost certainly will not merit testing in the low-risk population.

Taking on board your comments we have amended the background to ensure the focus population of this feasibility study remains clear, the asymptomatic high-risk population.

1.2. Methods

- Line 29: remove “non-invasive”

Author response: Amendment made as suggested (line 40).

1.3. Discussion

- Line 41: contrast cervical stiffness as a novel screening tool with the current screening tools (CL, fFN).

- Line 42: the authors may want to re-evaluate whether exploration of combining cervical stiffness, CL and fFN, into a multivariable model (cf. synergistic potential) is possible within the sample size.

Author response; Amendments made to discussion to ensure clarity in feasibility nature of the study. All references to fFN have been removed due to the unexpected removal of the product from the market by Hologic.

2. Introduction:

- Lines 50-59: In the first paragraph, the authors are providing some background to the PRECISION study. For the non-PTB researcher, the following would be helpful:

-reiteration of the clinical need (amendments line 62-69)

-the reasons why effective PTB risk prediction remains elusive (amendments line 70-74)

-mention of other predictors than cervical remodelling parameters like intrauterine infection and the role of FFN (amendments line 75-79)

-clarification that currently cervical remodelling assessment is based on sonographic assessments: CL (standard

care) and cervical consistency Index (subject to research) (amendments line 76-81 & 84-89)

-better explain why CCI has limitations (like?). It is noted that this statement appears at odds with the

statement made on page 13 of the Clinical Protocol, which mentions that CCI is an “easily, reproducible

assessment and effective in the prediction of spontaneous PTB”.(amendments line 89-92 “application of these proposed cervical consistency assessment tools relies upon expert ultrasound skills using expensive resources and no objective technique to assess cervical softness during pregnancy has so far been well established in clinical practice”)

- Line 66: The predictor under investigation is better defined as the Cervical Stiffness as obtained with the Pregnolia device and expressed as the CSI (mbar).(amendment line 99)

- Line 66: Within the context of PRECISION, it is unclear whether CSI is evaluated as a diagnostic biomarker (to confirm preterm labour; if yes – discussion should discuss comparison with QUIPP app decision support tool) or as a risk biomarker; it would be helpful if the authors can clarify this ambiguity, f.i, using the publicly available FDA-NIH resource on different types of biomarkers [https://www.ncbi.nlm.nih.gov/books/NBK338449/] (amendment line 99- “risk biomarker”)

- Lines 70-71: add reference to clinical guidance [NICE?] containing current gold standard practice as referred to (reference added line 107)

- Lines 70-73: The authors appear to suggest the use of longitudinal data for comparing CSI with CL and fFN; it is noted that such analyses are not mentioned in the statistical analysis? (statistical analysis has been amended to address this and to reflect feasibility outcomes from this study)

- Lines 73-74: As it stands, it is unclear which and how data from the PRECISION study will be used to inform a future clinical trial. It would be helpful if the authors can add this detail in the Discussion. (amendment line 104-110 and further clarification in feasibility outcomes statistical analysis)

- Note: The Background Section 1.1 in the actual Study Protocol is very well organized and clearly presented. The authors may consider replicating the same background presentation sequence (albeit condensed)

Author response: Thank you for your comments. We have amended the introduction to give a full background to the non-PTB researcher as to the relevance and importance of this research.

We have addressed all the line suggestions as indicated above and made amendments accordingly.

3. Materials & Methods

3.1. Study Design

- Line 79: The language suggests that recruitment occurs at first clinical appointment which happens between 14+0

– 25+6 weeks? Please clarify that recruitment is at first clinical appointment in the PTBC [14+0 – 17+6 weeks],

with 2 additional CS analyses scheduled as part of routine appointments scheduled before 25+6 weeks?

It is noted that the 3rd text box from the top in Figure 1 contains the same ambiguity.

- Line 84: If possible, reference guidance which outlines routine PTBC care

- Line 85: clarify that the questionnaire is only offered post first CL assessment.

Author response: Patients can be recruited at any gestation in the PTBC from 14-25+6 depending on when they attend clinic. Patients who are recruited at study visit A will be between 14-17+6 weeks gestation, we would then hope to obtain further CS assessments at their subsequent clinic appointments that will correspond with Study visit B (18-21+6 weeks) and Study visit C (22+0-25+6). However, if a patient is first recruited when they attend clinic at 18+2 weeks, their study procedures at that visit will correspond to study visit B, they will have already missed study visit A.

This study design allows for patients who may miss their first appointment or who have been referred to clinic late and gives them the opportunity to take part in the study even if they miss time point study visit A. The ability to determine which study visit timepoint achieves the best recruitment will inform future study design.

Amendments made to figure 1.

All guidelines are referenced at the end of the paragraph on line 127.

We confirm that the questionnaire is only offered at the participants first study visit (can be A,B or C depending on their gestation). Figure 1 has been amended to make this clear.

3.2. Study Objective

- Line 106: the language in objective 1.3 is unclear, can the authors clarify what is meant with “… explore the relative association with gestation at delivery”.

- Line 109: Objective 3 – details on how this study will inform a next study would be helpful; see higher.

Author response: As a feasibility study, one of the aims is to search for any possible associations that may be worth following u

---

## [Decision Letter · Decision Letter 1]

8 Oct 2024

PONE-D-24-18823R1Study Protocol. The PRECISION study protocol. Can cervical stiffness in the second trimester predict preterm birth in high-risk singleton pregnancies? A feasibility, cohort study.PLOS ONE

Dear Dr. Medford,

Thank you for submitting your manuscript to PLOS ONE. After careful consideration, we feel that it has merit but does not fully meet PLOS ONE’s publication criteria as it currently stands. Therefore, we invite you to submit a revised version of the manuscript that addresses the points raised during the review process.

**ACADEMIC EDITOR: Please respond to all reviewers comments**

We look forward to receiving your revised manuscript.

Kind regards,

Ahmed Mohamed Maged, MD

Academic Editor

PLOS ONE

**Comments from PLOS Editorial Office:** We note that one or more reviewers has recommended that you cite specific previously published works in an earlier round of revision. As always, we recommend that you please review and evaluate the requested works to determine whether they are relevant and should be cited. It is not a requirement to cite these works and you may remove them before the manuscript proceeds to publication. We appreciate your attention to this request.

Reviewers' comments:

Reviewer's Responses to Questions

**Comments to the Author**

1. Does the manuscript provide a valid rationale for the proposed study, with clearly identified and justified research questions?

Reviewer #2: Yes

2. Is the protocol technically sound and planned in a manner that will lead to a meaningful outcome and allow testing the stated hypotheses?

Reviewer #2: Yes

3. Is the methodology feasible and described in sufficient detail to allow the work to be replicable?

Reviewer #2: Yes

4. Have the authors described where all data underlying the findings will be made available when the study is complete?

Reviewer #2: Yes

5. Is the manuscript presented in an intelligible fashion and written in standard English?

Reviewer #2: Yes

6. Review Comments to the Author

You may also provide optional suggestions and comments to authors that they might find helpful in planning their study.

Reviewer #2: SUMMARY

In their revised manuscript on the PRECISION study protocol, Medford et al outline a single-centre prospective cohort feasibility study involving pregnant women identified to be at high risk of spontaneous preterm birth which aims to evaluate cervical stiffness as a predictor of preterm birth at different 3 timepoints in pregnancy. Cervical Stiffness is measured using a novel medical device, “Pregnolia” which measures the vacuum as required to deform cervix tissue to a certain extent. The vacuum required to obtain the deformation provides a measure of cervical stiffness (high vacuum required) or cervical softening (low vacuum required), with the latter indicative for cervical remodelling preceding cervical shortening and possible preterm labour. In addition, the authors want to compare Cervical Stiffness with Cervical Length analyses for Preterm birth (before 34 weeks of gestation) prediction using ROC analysis and association with gestational age at delivery. Comparison with fetal fibronectin has been removed, following market withdrawal of this test. The PRECISION study will collect patient feedback on how they experience Cervical Stiffness assessments. The outputs of the PRECISION feasibility study will be used to inform a possible follow-up clinical study

ADDRESSING THE MAJOR ISSUES IDENTIFIED

In their revised manuscript, Medford et al addressed to a satisfactory extent the major issues identified in their original submission.

In their revised manuscript, the authors provided an acceptable justification for their sample size and thus the ability for any reader to make up their mind on the strength of the projected research findings.

The authors also significantly improved the clarity of their research aims and planned analyses.

Overall, the authors took many of the reviewer comments in account in a positive fashion and modified their manuscript accordingly. This willingness to address critique constructively stands to the authors.

It is noted that in their response to the “Major Issue”, the authors justify the PRECISION feasibility study in the context of the larger Breuking study by highlighting the latter assumes a standard level of variance to inform their power calculation. This justification is conducive to the reviewer. However, variance analysis is not (explicitly) part of the statistical analysis plan. The authors are invited to amend their SAP accordingly.

At the same time, it is advised to briefly explain the appropriateness of Cronbach’s alpha (used to e.g. assess science education “instruments”) in the context of replicate measurements of a vacuum pressure to assess tissue stiffness. Classic estimations of variance (standard deviation, %CV) etc may be more appropriate. Typically, triplicate measurements are used to generate a more precise estimate (mean) given that each measuring procedure has some variance. [triplicate measurements are however not ideal to estimate the within-run imprecision directly]. In their response to the “Major Issue” and the SAP, the authors appear to suggest that there may be other (biological) effects at play (other than plain measurement imprecision); is this the reason why Cronbach’s alpha is proposed to gauge measurement reliability? This warrants some further explanation, e.g. in the discussion.

MINOR ISSUES

1. Abstract

1.1. Background

1.1.1. Line 29: Suggested rephrasing: “Therefore, existing care pathways for managing PTB risk can potentially benefit… “

1.2. Methods

1.2.1. Line 37: Suggestion to replace “All women” by “All study participants….”

1.2.2. Line 45: Suggestion to replace “This assessment….” By “The cervical stiffness index data will be…”

1.3. Discussion

1.3.1. Line 50: Suggested Rephrasing: This is an exploratory study to assess the use of…

2. Introduction:

2.1.1. Line 64: Suggested Rephrasing: Unfortunately, it remains…

2.1.2. Line 70: Suggested Rephrasing: Understanding the events leading to sPTB remains

2.1.3. Line 74: Replace “researching” by “predicting”

2.1.4. Line 91: “…expensive resources. So far, no…”

2.1.5. Line 97: Remove “ongoing”

2.1.6. Line 101: Suggestion to Replace “explore” by “investigate”

2.1.7. Line 102: Suggestion to Replace “relevant” by “representative” & “ensure” by “justify a”

2.1.8. Line 103: Suggested Rephrasing: ….predictive study and inform its appropriate design and power.

2.1.9. Line 104: Suggestion to Replace “explore if” by “confirm whether”

2.1.10. Line 106: Add “also”: The study will also…

2.1.11. Line 109. Move/adopt the following language from Section “Statistical Analysis- sample size” to here: … in these patients. “ In addition, the impact of the introduction of the Saving Babies Lives Care Bundle v3 on patient participation and local PTB rates in the asymptomatic high PTB risk pregnancy population will be determined.”

3. Materials & Methods

3.1. Study Design

3.1.1. Line 117: Suggested Rephrasing: “…during their PTB surveillance at the…”

3.1.2. Line 118: Suggested Rephrasing: “The CSI measurements will be taken alongside…” If possible, reference guidance which outlines routine PTBC care

3.1.3. Line 123: Suggested to move/modify language lines 130-131 to Line 123, e.g.: “Participants attending at the Study Visit A time window will be invited to consent to gifting… etc”

3.1.4. Line 124: Suggestion to Replace “coordinate” by “coincide”

3.2. Study Objective

3.2.1. Line 146: Suggested Rephrasing: “Upon positive conclusion of the PRECISION feasibility study, to inform the design…”

3.3. Study Population

3.3.1. Line 151: Suggested Rephrasing: “The Precision study population… “

3.3.2. Line 153: Replace “Patient’s fulfilling” by “Patients meeting”

3.4. Inclusion and Exclusion Criteria:

3.4.1. Line 169: Spell out “LLETZ”

4. Study Procedures

4.1. Pregnolia System

4.1.1. Line 191: Suggested Rephrasing: “The Pregnolia system provides audio guidance to the clinician by indicating….

4.1.2. Line 191: Add: clear “operation” instructions

4.2. Cervical Stiffness Assessment

4.2.1. OK

4.3. Cervical Length Assessment

4.3.1. OK

4.4. Table 1:

4.4.1. Remove the row with row label: “Fetal fibronectin swab collection” – unless these were collected in patients already enrolled. If so, add an asterisk with a brief note that this procedure was discontinued at Date following removal of fetal fibronectin test from UK market by manufacturer.

5. Study Outcome

5.1. Feasibility outcomes

5.1.1. Line 257: add the aim metric (80% of ????) as Triplicate measurement reliability is not a common metric in the context of taking repeat pressure measurements

5.1.2. It is suggested to add explicit language regarding inferring inputs for power calculation of a future study and introduce a feasibility outcome

5.1.2.1. regarding collecting of sPTB rates in the total study population (participating and non-participating PTBC patients) to inform sPTB rate post care bundle v3 implementation – remove the language from section Statistical Analysis – Sample Size

5.1.2.2. regarding CSI measurement variance in this population to inform power calculation; remove the language from section Statistical Analysis – Sample Size

5.1.2.3. Regarding recruitment and participation rates to inform number of participants needed ; remove the language from section Statistical Analysis – Sample Size

5.2. Clinical Outcomes

5.2.1. Line 261: Suggested Rephrasing: “The primary clinical outcome is…”

5.2.2. Line 263: Spell out “PPROM”

6. Study Endpoint

6.1.1. OK

7. Statistical Analysis

7.1.1. Move lines 273-276 to precede subsection ”sample size”

7.2. Sample Size

7.2.1. Make paragraph more concise; some of “explanation” language can be moved to introduction see point 2.1.11. Suggested Rephrasing: “ From previous studies at this study site, an 18% PTB rate was found for the local asymptomatic high risk population; equally these studies yielded acceptable recruitment / participation rates. Following the recent introduction of the Saving Babies Lives Care Bundle v3, which changes clinical practice, these rates need to be re-established. In this feasibility study…”

7.2.2. When the authors amend the Feasibility outcomes as per suggestion 5.1.2, then the lines 284-288 can be removed.

7.3. Statistical Analysis

7.3.1. Lines 297-303: Mention that CL comparator diagnostic performance data will also be generated.

8. Data Management Plan

8.1. OK

9. Consent and criterial for withdrawal

9.1. OK

10. Monitoring and Safety

10.1. OK

11. Ethical Considerations and declarations

11.1. OK

12. Status and Timeline of the Study

12.1. OK

13. Discussion

13.1. Line 372 – 373: Consider Rephrasing the concluding words: “…vital change in such a devastating cohort of obstetrics” is an incorrect phrase?

7. PLOS authors have the option to publish the peer review history of their article (what does this mean? ). If published, this will include your full peer review and any attached files.

**Do you want your identity to be public for this peer review?** For information about this choice, including consent withdrawal, please see our Privacy Policy .

Reviewer #2: No

---

## [Author Response · Author response to Decision Letter 2]

19 Nov 2024

Comments from PLOS Editorial Office: We note that one or more reviewers has recommended that you cite specific previously published works in an earlier round of revision. As always, we recommend that you please review and evaluate the requested works to determine whether they are relevant and should be cited. It is not a requirement to cite these works and you may remove them before the manuscript proceeds to publication. We appreciate your attention to this request.

Author response: These previously published works referred to a prediction analysis that we are unable to do, as this is a feasibility study, therefore we have not cited them in this paper.

ADDRESSING THE MAJOR ISSUES IDENTIFIED

It is noted that in their response to the “Major Issue”, the authors justify the PRECISION feasibility study in the context of the larger Breuking study by highlighting the latter assumes a standard level of variance to inform their power calculation. This justification is conducive to the reviewer. However, variance analysis is not (explicitly) part of the statistical analysis plan. The authors are invited to amend their SAP accordingly.

At the same time, it is advised to briefly explain the appropriateness of Cronbach’s alpha (used to e.g. assess science education “instruments”) in the context of replicate measurements of a vacuum pressure to assess tissue stiffness. Classic estimations of variance (standard deviation, %CV) etc may be more appropriate. Typically, triplicate measurements are used to generate a more precise estimate (mean) given that each measuring procedure has some variance. [triplicate measurements are however not ideal to estimate the within-run imprecision directly]. In their response to the “Major Issue” and the SAP, the authors appear to suggest that there may be other (biological) effects at play (other than plain measurement imprecision); is this the reason why Cronbach’s alpha is proposed to gauge measurement reliability? This warrants some further explanation, e.g. in the discussion.

Author response:

In this feasibility study we will obtain an estimate of the variability in the outcome measure which can then be included in a sample size calculation for the full definitive study. I have made an amendment to the statistical analysis plan to clarify this. Lines 299.

Triplicate measurements have been taken as per the manufacturer guidance in using the Pregnolia System. The cervical tissue has natural elasticity and takes up to 30mins to return to its normal tissue structure after 1 reading, therefore when obtaining 3 readings in quick succession (as instructed to do so) there is expected to be a difference in the stiffness readings obtained, the first being the highest with the most pressure required to deform the tissue by 4mm, and as the tissue will not yet have returned to its natural state the 2nd and 3rd readings will be expected to require less pressure each time. Despite being informed by the manufacturer to take 3 successive readings, they have not advised on how to interpret or use those readings for clinical practice. Determining the reliability of the device readings will inform how to interpret and use these results in real clinical practice. This is referred to in the Discussion (line 367) and with the amendments made to the SAP the use of Cronbach’s alpha to achieve this should be clearer (line 292).

According to our statistician, determining Cronbach’s Alpha assesses the reliability of the device to produce consistent readings at different time points. It compares shared variability, among the measurements with the overall variance. If the device is reliable then there should be high covariance amongst the measurements relative to the variance. Added to statistical analysis plan. (line 292).

MINOR ISSUES

1. Abstract

1.1. Background

1.1.1. Line 29: Suggested rephrasing: “Therefore, existing care pathways for managing PTB risk can potentially benefit… “ Line 29 Amended as suggested.

1.2. Methods

1.2.1. Line 37: Suggestion to replace “All women” by “All study participants….” Line 37 amended as suggested.

1.2.2. Line 45: Suggestion to replace “This assessment….” By “The cervical stiffness index data will be…”Line 45 amended as suggested.

1.3. Discussion

1.3.1. Line 50: Suggested Rephrasing: This is an exploratory study to assess the use of…Line 50 amended as suggested.

2. Introduction:

2.1.1. Line 64: Suggested Rephrasing: Unfortunately, it remains…Line 63 amended as suggested.

2.1.2. Line 70: Suggested Rephrasing: Understanding the events leading to sPTB remains…Line 70 amended as suggested.

2.1.3. Line 74: Replace “researching” by “predicting” Line 75 amended as suggested.

2.1.4. Line 91: “…expensive resources. So far, no…”Line 92 amended as suggested.

2.1.5. Line 97: Remove “ongoing” Line 98 amended as suggested.

2.1.6. Line 101: Suggestion to Replace “explore” by “investigate” Line 102 amended as suggested.

2.1.7. Line 102: Suggestion to Replace “relevant” by “representative” & “ensure” by “justify a” Line 103 amended as suggested.

2.1.8. Line 103: Suggested Rephrasing: ….predictive study and inform its appropriate design and power. line 104 amended as suggested.

2.1.9. Line 104: Suggestion to Replace “explore if” by “confirm whether”. Line 105 amended as suggested.

2.1.10. Line 106: Add “also”: The study will also… Line 107 amended as suggested.

2.1.11. Line 109. Move/adopt the following language from Section “Statistical Analysis- sample size” to here: … in these patients. “ In addition, the impact of the introduction of the Saving Babies Lives Care Bundle v3 on patient participation and local PTB rates in the asymptomatic high PTB risk pregnancy population will be determined.” Amendments made, but moved to study outcomes rather than introduction. Lines 235-248.

3. Materials & Methods

3.1. Study Design

3.1.1. Line 117: Suggested Rephrasing: “…during their PTB surveillance at the…” line 119 amended as suggested

3.1.2. Line 118: Suggested Rephrasing: “The CSI measurements will be taken alongside…” If possible, reference guidance which outlines routine PTBC care Line 130 amended as suggested and reference added.

3.1.3. Line 123: Suggested to move/modify language lines 130-131 to Line 123, e.g.: “Participants attending at the Study Visit A time window will be invited to consent to gifting… etc” Line 125 amended as suggested.

3.1.4. Line 124: Suggestion to Replace “coordinate” by “coincide” Amended as suggested, line 128.

3.2. Study Objective

3.2.1. Line 146: Suggested Rephrasing: “Upon positive conclusion of the PRECISION feasibility study, to inform the design…” Line 150 amended as suggested.

3.3. Study Population

3.3.1. Line 151: Suggested Rephrasing: “The Precision study population… “ Line 155 amended as suggested

3.3.2. Line 153: Replace “Patient’s fulfilling” by “Patients meeting” line 157 amended as suggested

3.4. Inclusion and Exclusion Criteria:

3.4.1. Line 169: Spell out “LLETZ” line 173 amended as suggested

4. Study Procedures

4.1. Pregnolia System

4.1.1. Line 191: Suggested Rephrasing: “The Pregnolia system provides audio guidance to the clinician by indicating….Line 195 amended as suggested.

4.1.2. Line 191: Add: clear “operation” instructions Line 197 amended as suggested.

4.2. Cervical Stiffness Assessment

4.2.1. OK

4.3. Cervical Length Assessment

4.3.1. OK

4.4. Table 1:

4.4.1. Remove the row with row label: “Fetal fibronectin swab collection” – unless these were collected in patients already enrolled. If so, add an asterisk with a brief note that this procedure was discontinued at Date following removal of fetal fibronectin test from UK market by manufacturer. Amended as suggested.

5. Study Outcome

5.1. Feasibility outcomes

5.1.1. Line 257: add the aim metric (80% of ????) as Triplicate measurement reliability is not a common metric in the context of taking repeat pressure measurements. Many thanks for highlighting this error. Cronbach’s alpha metric greater than 0.8 is considered to have good consistency, and so amendment from 80% to 0.8, line 269.

5.1.2. It is suggested to add explicit language regarding inferring inputs for power calculation of a future study and introduce a feasibility outcome. Suggestion noted, added at line 270.

5.1.2.1. regarding collecting of sPTB rates in the total study population (participating and non-participating PTBC patients) to inform sPTB rate post care bundle v3 implementation – remove the language from section Statistical Analysis – Sample Size. Removed from sample size as suggested and added line 271.

5.1.2.2. regarding CSI measurement variance in this population to inform power calculation; remove the language from section Statistical Analysis – Sample Size. Removed from sample size and added as line 274.

5.1.2.3. Regarding recruitment and participation rates to inform number of participants needed ; remove the language from section Statistical Analysis – Sample Size Removed from sample size and added as line 275.

5.2. Clinical Outcomes

5.2.1. Line 261: Suggested Rephrasing: “The primary clinical outcome is…”line 279 amended as suggested

5.2.2. Line 263: Spell out “PPROM” abbreviation added at line 166

6. Study Endpoint

6.1.1. OK

7. Statistical Analysis

7.1.1. Move lines 273-276 to precede subsection ”sample size” Moved to study outcome section, line 235-238.

7.2. Sample Size

7.2.1. Make paragraph more concise; some of “explanation” language can be moved to introduction see point 2.1.11. Suggested Rephrasing: “ From previous studies at this study site, an 18% PTB rate was found for the local asymptomatic high risk population; equally these studies yielded acceptable recruitment / participation rates. Following the recent introduction of the Saving Babies Lives Care Bundle v3, which changes clinical practice, these rates need to be re-established. In this feasibility study…” Suggestions noted, and amendments made moving section to study design (line 144), study outcomes line (239-248) and feasibility outcomes (line 271).

7.2.2. When the authors amend the Feasibility outcomes as per suggestion 5.1.2, then the lines 284-288 can be removed. Removed as suggested.

7.3. Statistical Analysis

7.3.1. Lines 297-303: Mention that CL comparator diagnostic performance data will also be generated. Line 306 amended as suggested

8. Data Management Plan

8.1. OK

9. Consent and criterial for withdrawal

9.1. OK

10. Monitoring and Safety

10.1. OK

11. Ethical Considerations and declarations

11.1. OK

12. Status and Timeline of the Study

12.1. OK

13. Discussion

13.1. Line 372 – 373: Consider Rephrasing the concluding words: “…vital change in such a devastating cohort of obstetrics” is an incorrect phrase? Amendment made to line 380.

---

## [Decision Letter · Decision Letter 2]

10 Dec 2024

Study Protocol. The PRECISION study protocol. Can cervical stiffness in the second trimester predict preterm birth in high-risk singleton pregnancies? A feasibility, cohort study.

PONE-D-24-18823R2

Dear Dr. Medford,

We’re pleased to inform you that your manuscript has been judged scientifically suitable for publication and will be formally accepted for publication once it meets all outstanding technical requirements.

Kind regards,

Ahmed Mohamed Maged, MD

Academic Editor

PLOS ONE

Additional Editor Comments (optional):

Reviewers' comments:

Reviewer's Responses to Questions

**Comments to the Author**

1. Does the manuscript provide a valid rationale for the proposed study, with clearly identified and justified research questions?

Reviewer #2: Yes

2. Is the protocol technically sound and planned in a manner that will lead to a meaningful outcome and allow testing the stated hypotheses?

Reviewer #2: Yes

3. Is the methodology feasible and described in sufficient detail to allow the work to be replicable?

Reviewer #2: Yes

4. Have the authors described where all data underlying the findings will be made available when the study is complete?

Reviewer #2: Yes

5. Is the manuscript presented in an intelligible fashion and written in standard English?

Reviewer #2: Yes

6. Review Comments to the Author

You may also provide optional suggestions and comments to authors that they might find helpful in planning their study.

Reviewer #2: SUMMARY

In their revised manuscript on the PRECISION study protocol, Medford et al outline a single-centre prospective cohort feasibility study involving pregnant women identified to be at high risk of spontaneous preterm birth which aims to evaluate cervical stiffness as a predictor of preterm birth at different 3 timepoints in pregnancy. Cervical Stiffness is measured using a novel medical device, “Pregnolia” which measures the vacuum as required to deform cervix tissue to a certain extent. The vacuum required to obtain the deformation provides a measure of cervical stiffness (high vacuum required) or cervical softening (low vacuum required), with the latter indicative for cervical remodelling preceding cervical shortening and possible preterm labour. In addition, the authors want to compare Cervical Stiffness with Cervical Length analyses for Preterm birth (before 34 weeks of gestation) prediction using ROC analysis and association with gestational age at delivery. The PRECISION study will collect patient feedback on how they experience Cervical Stiffness assessments. The outputs of the PRECISION feasibility study will be used to inform a possible follow-up clinical study.

ADDRESSING THE MAJOR ISSUES IDENTIFIED

In their revised manuscript, Medford et al addressed adequately addressed all the reviewer’s comments.

MINOR ISSUES

1. Line 93: remove “so far”

2. Line 196: replace… “to the clinician by indicating when the measurement has started, in progress and completed.” By ““to the clinician by indicating that the measurement has started, is in progress, or was completed.”

3. Line 244: replace “populations” by “population’s”

4. Line 247: Replace “Together this data can infer larger study design for a predictive model in

5. this population.” By “Together these data will be used to infer the design of a larger study to establish a predictive model for spontaneous preterm birth (<34 weeks) in this patient population”

6. Line 330: Replace “date by “data”

7. PLOS authors have the option to publish the peer review history of their article (what does this mean? ). If published, this will include your full peer review and any attached files.

**Do you want your identity to be public for this peer review?** For information about this choice, including consent withdrawal, please see our Privacy Policy .

Reviewer #2: No

---

## [Editor Report · Acceptance letter]

PONE-D-24-18823R2

PLOS ONE

Dear Dr. Medford,

I'm pleased to inform you that your manuscript has been deemed suitable for publication in PLOS ONE. Congratulations! Your manuscript is now being handed over to our production team.

Kind regards,

on behalf of

Professor Ahmed Mohamed Maged

Academic Editor

PLOS ONE